# State of the Art on the SARS-CoV-2 Toolkit for Antigen Detection: One Year Later

**DOI:** 10.3390/bios11090310

**Published:** 2021-08-31

**Authors:** Laura Fabiani, Veronica Caratelli, Luca Fiore, Viviana Scognamiglio, Amina Antonacci, Silvia Fillo, Riccardo De Santis, Anella Monte, Manfredo Bortone, Danila Moscone, Florigio Lista, Fabiana Arduini

**Affiliations:** 1Department of Chemical Science and Technologies, University of Rome “Tor Vergata”, Via della Ricerca Scientifica, 00133 Rome, Italy; laura.fabiani@uniroma2.it (L.F.); veronica.caratelli@uniroma2.it (V.C.); luca.fiore@uniroma2.it (L.F.); moscone@uniroma2.it (D.M.); 2Institute of Crystallography (IC-CNR), Department of Chemical Sciences and Materials Technologies, Via Salaria km 29.300, 00015 Monterotondo, Italy; viviana.scognamiglio@ic.cnr.it (V.S.); amina.antonacci@ic.cnr.it (A.A.); 3Scientific Department, Army Medical Center, Via Santo Stefano Rotondo 4, 00184 Rome, Italy; silviafillo@gmail.com (S.F.); riccardo.desantis@gmail.com (R.D.S.); nellymonte88@gmail.com (A.M.); fabiana_arduini@yahoo.it (M.B.); romano.lista@gmail.com (F.L.); 4SENSE4MED, via Renato Rascel 30, 00128 Rome, Italy

**Keywords:** SARS-CoV-2 detection, nasopharyngeal swab, saliva, serum, droplets

## Abstract

The recent global events of COVID-19 in 2020 have alerted the world to the risk of viruses and their impacts on human health, including their impacts in the social and economic sectors. Rapid tests are urgently required to enable antigen detection and thus to facilitate rapid and simple evaluations of contagious individuals, with the overriding goal to delimitate spread of the virus among the population. Many efforts have been achieved in recent months through the realization of novel diagnostic tools for rapid, affordable, and accurate analysis, thereby enabling prompt responses to the pandemic infection. This review reports the latest results on electrochemical and optical biosensors realized for the specific detection of SARS-CoV-2 antigens, thus providing an overview of the available diagnostics tested and marketed for SARS-CoV-2 antigens as well as their pros and cons.

## 1. Introduction

Since the early 1900s, diverse pathogenic viruses have been identified to cause severe diseases worldwide, including Rift Valley fever (1931), Crimean Congo hemorrhagic fever (1944), Zika virus (1947), Chikungunya (1952), Marburg (1967), Lassa fever (1969), Ebola virus (1976), human immunodeficiency virus (1980), Nipah (1998), severe acute respiratory syndrome (SARS, 2003), influenza A virus (2009), and Middle East respiratory syndrome coronavirus (MERS, 2012), among others (Figure 1A) [1]. In 2020, severe acute respiratory syndrome-coronavirus (SARS-CoV-2) caused an outbreak of the respiratory disease named COVID-19, which has had a significant impact on human health and in all economic sectors [2], leading to the most serious socioeconomic crisis since World War II [3].

Initially, the diagnosis of COVID-19 was usually carried out in the hospital by medical imaging through high-cost instrumentation and skilled personnel, including the use of computed tomography, radiograph X-rays, ultrasound, echocardiograms, and magnetic resonance imaging (Figure 1B) [4]. However, considering the wide spread of COVID-19, the availability of a cost-effective and laboratory-free detection method would help to prevent outbreaks of the virus and lessen its associated mortality.

In general, SARS-CoV-2 detection systems are divided into three general categories: (i) RNA-based diagnostics, (ii) antigen-based diagnostics, and (iii) antibody-based diagnostics (Figure 1C) [5].

To establish a unique approach in the application of the available diagnostics, the European Commission (EU) released “Guidelines on COVID-19 for in vitro diagnostic tests and their performance (2020/C 122 I/01)”. The aim of this publication was to outline the regulatory context of the in vitro diagnostic devices used in EU countries and provide an overview of the different procedures and purposes of these tests. In detail, the tests available today for COVID-19 fall broadly into two categories: (i) tests based on evaluating the contagiousness of SARS-CoV-2 through the detection of viral genetic material (Polymerase Chain Reaction) and viral components, such as proteins on the virus surface (antigen tests); and (ii) tests that estimate exposure to the virus based on the immune response of the human body to the infection. However, the EU also highlighted that diagnostics for immune response have been not able to provide “a definite answer on the presence or absence of the SARS-CoV-2 virus and thus they are not suitable to assess if the tested individual may be contagious for others. Nevertheless, antibody tests could prove essential for performing large-scale sero-epidemiological population surveys for assessing, e.g., the immune status of workers and as one of the elements for guiding de-escalation strategies when the pandemic is under control” [6].

For this reason, rapid tests are urgently required for antigen detection to easily and quickly evaluate contagious individuals and thus delimitate spread of the virus among the population. To this end, on 30 January 2020, the EC promptly launched a Call for Projects entitled “SC1-PHECORONAVIRUS-2020: Advancing knowledge for the clinical and public health response to the [COVID-19] epidemic”, featuring 18 Projects with a budget of EUR 48.5 million and involving 151 research groups across Europe and beyond, for research activities devoted to counteracting the COVID-19 emergency [7]. Four main pillars have been proposed based on: (i) infection monitoring systems, (ii) point-of-care diagnostic tests, (iii) new treatments, and (iv) the development of new vaccines. Among them, the requirement for novel rapid diagnostics “will concentrate on enabling front-line health workers to make the diagnosis more quickly and more accurately, which will, in turn, reduce the risk of further spread of the virus”, according to the EC.

The following projects have been awarded support for the development of novel diagnostics:-CoNVat: Combating 2019-nCoV: Advanced Nanobiosensing platforms for POC global diagnostics and surveillance to develop a point-of-care device using optical biosensor technology for rapid diagnosis and monitoring of the new coronavirus directly in the patient’s sample (four partners: ES(2), FR, and IT) [8].-CORONADX: Three rapid diagnostic tests (point-of-care) for COVID-19 Coronavirus, improving epidemic preparedness, public health, and socioeconomic benefits to deliver three complementary diagnostic tools, including one point-of-care diagnostic that can be used with minimal training (eight partners: AT, CN(2), DK(2), IT(2), and SE) [9].-HG nCoV19 test: Development and validation of a rapid molecular diagnostic test for nCoV19 to develop and validate a novel rapid molecular diagnostic test for coronavirus (four partners: CN, IE, IT, and UK) [10].

It is evident that the high potential of diagnostics could help improve knowledge on virus diffusion, as well as diminish the danger of further spread, considering the continuous emergence and re-emergence of viral infections, as highlighted by Cheng et al. [11]. As the authors asserted, “The findings that horseshoe bats are the natural reservoir for SARS-CoV-like virus and that civet are the amplification host highlight the importance of wildlife and biosecurity in farms and wet markets, which can serve as the source and amplification centers for emerging infections”.

Therefore, early-stage detection of viral infection could help to circumvent further infection from highly contagious viruses and prevent viral disease morbidity and premature death among the worldwide population. To highlight the relevance of biosensors as smart analytical tools, several authors reviewed the advantages and disadvantages of the many biosensing configurations realized for the detection of viruses [12,13,14]. Eden Morales-Narváez and Can Dincer discussed the potential of using biosensing tools beyond PCR-based systems, reporting that “researchers around the world are pushing hard to develop different methods and devices, allowing an easy, rapid, affordable and highly sensitive and selective quantification of nucleic acids in low resource settings (such as doctors’ practices, or directly at home)” [15]. This review, to our knowledge, was the first on the topic of biosensors and COVID-19, and its impact as a widely cited paper was recently recognized by Clarivate Web of Science (May 2021). Afterward, other reviews were published, highlighting how nanotechnology [13,14,15] plays a crucial role in the design of reliable and miniaturized devices.

The lessons learned during the first year of the COVID-19 pandemic guided the design of innovative toolkits with strong potential to be exploited for wide screening under low resource settings. As outlined by Bhalla et al. [16], an ideal biosensor for effective use in pandemics should be single-use and offer a long shelf life, ease of use, cost-effectiveness, mass-manufacturing ability, autonomy, high sensitivity, high selectivity, rapidity, multiplexing capabilities, and multiple sensing modes (Figure 1D). The convergence of interdisciplinary technologies represents an immediate solution for the main bottlenecks constraining biosensor prototypes to achieve real applications with the desirable features. Such technologies include: (i) nanomaterial technology to improve the analytical figures of merit; (ii) microfluidics to enhance biosensor performance in terms of sustainability (by decreasing the use of reagents and waste volume), automation, and suitability for in-field analysis; (iii) smartphone-assisted technology to support the realization of easy-to-use and portable systems, thus boosting data transmission and management in a timely fashion; and (iv) wearable tools to help collect previously inaccessible physical and biochemical signals, including those from epidermal tattoos, contact lenses, textiles, face masks, wristbands, and patches [17].

This review provides an overview of the biosensors for SARS-CoV-2 antigens available in the literature and tested on real matrices, such as nasopharynx swabs, saliva, serum, and droplets, approximately one year after the start of the COVID-19 outbreak. Indeed, beyond the development of biosensors for RNA sequences and antibodies, the detection of antigens such as the SARS-CoV-2 spike (S) protein and SARS-CoV-2 nucleocapsid (N) protein by biosensing tools has attracted significant attention. The pros and cons for the detection of S and N proteins are also highlighted to develop a feasible strategy for fabricating an ideal biosensor for the specific detection of SARS-CoV-2, thereby overcoming the limitation of commercially available lateral-flow immunosensors (Figure 2 and Table 1), which encompass the use of invasive nasopharyngeal swabs as the sampling system, which offers lower sensitivity.

## 2. SARS-CoV-2 Antigen Detection Using Nasopharyngeal Swab

The first biosensor described in the literature for detecting the SARS-CoV-2 antigen was developed by the Seo et al. [26]. The method involves using the developed biosensor to measure S protein in nasopharyngeal swab specimens (Figure 3A).

The S protein was selected because it is a superficial glycoprotein of SARS-CoV-2 with an affinity for human angiotensin-converting enzyme 2 (hACE2), which is used as a receptor to infect human cells [27,28]. Nasopharyngeal swab specimens were selected because they offer the highest-yield samples for diagnostic testing, even though the collection of such samples, while generally considered safe, is invasive [29]. In detail, the Seo research group developed a graphene-based field-effect transistor immunosensor by immobilizing the SARS-CoV-2 spike antibody through the 1-pyrenebutyric acid N-hydroxysuccinimide ester, enabling detection of the S protein with a detection limit of 1 fg/mL in a standard solution. This field-effect transistor-based immunosensor was tested in nasopharyngeal swab specimens from COVID-19 patients and a cultured virus, observing detection limits equal to 100 fg/mL and 1.6 × 10^1^ pfu/mL, respectively. This article opened global research avenues for the development of immunosensor-based sensors for the S protein, demonstrating the suitability of the immunosensing system for the rapid detection of patients affected by COVID-19. Besides the S protein, the N protein of the coronavirus is often used as a marker in diagnostic assays. Other authors also highlighted the usefulness of the N antigen of SARS-CoV-2 for reliable diagnosis [30,31].

Shao et al. used the same field-effect transistor approach by employing high-purity semiconducting single-walled carbon nanotubes functionalized with specific antibodies for the detection of both S and N proteins. The detection of S and N proteins was carried out by adding 10 μL of a nasopharyngeal swab sample for 2 min [32]. After this incubation time, the device was washed three times with water, and the measurement performed. The two types of field-effect transistor immunosensors demonstrated a LOD of 0.55 fg/mL for the S protein and 0.016 fg/mL, respectively, for the N protein in a standard solution. To evaluate the feasibility in real samples, a total of 28 PCR-positive samples and 10 negative nasopharyngeal swab samples were tested, finding a 17.8% false-negative rate. The low detection achieved for both proteins alongside the technique’s application in clinical samples demonstrated the feasibility of immuno-based field-effect transistors for the rapid detection of SARS-CoV-2 in nasopharyngeal swab specimens. Furthermore, Chaimayo et al. proposed a rapid SARS-CoV-2 antigen detection test, the Standard™ Q COVID-19 Ag kit, which was able to detect SARS-CoV-2 in nasopharyngeal and throat swabs collected from 454 suspected COVID-19 cases. This Standard Q COVID-19 Ag test provides a rapid chromatographic immunoassay for the detection of N protein characterized by two precoated lines on the result window: a control (C) line coated with a mouse monoclonal anti-chicken Igγ antibody and a test (T) line coated with a mouse monoclonal antibody against N protein. The antigen–antibody color particles migrate via a complex process involving capillary force and are captured by the mouse monoclonal anti-SARS-CoV-2 antibody coated on the test (T) region. The colored test (T) line’s intensity depends on the amount of SARS-CoV-2 N antigen present in the sample. This rapid Standard™ Q COVID-19 Ag kit showed comparable sensitivity (98.33%; 95% CI, 91.06–99.96%) and specificity (98.73%; 95% CI, 97.06–99.59%) to a real-time RT-PCR assay, demonstrating its potential use as a screening assay, especially in high prevalence areas [33].

Although commercially available antibodies are used as the main approach for detecting SARS-CoV-2 antigens, Kim et al. developed single-chain variable fragment (scFv)-crystallizable (Fc) fusion proteins (scFv-Fcs) for the detection of N protein with improved specificity [34]. To screen scFv binders that specifically interact with the SARS-CoV-2 N protein, the authors carried out a phage-display screening using a chicken-naïve scFv antibody library, followed by the isolation of positive clones and the elimination of scFv binders from MERS-CoV and SARS-CoV-2. After producing the specific antibodies, an analysis was carried out using four unique clones, 12H1, 12H8, 12B3, and 1G5, all characterized by different complementary–determining region sequences for heavy and light chains. The sensitivity was evaluated by binding experiments using the SARS-CoV-2 N protein, finding K_D_ values equal to 18.3, 1.31, 8.47, and 2.86 nM, respectively. The lateral-flow assay was designed by introducing the scFv-Fc antibody on the test line and anti-human IgG antibodies on the control line, while the conjugate pad was loaded with each scFv-Fc antibody–cellulose nanobead conjugate. The analytical performance of the developed device was tested using 100 μL of N protein or cultured virus in a lysis buffer, with 20 min as the analysis time. Then, the LOD was evaluated and found to be 2, 5, and 10 ng when using 12H8–12H1, 12H8–12B3, and 12H8-1G5, respectively (Figure 3B). When tested with the cultured virus, a LOD equal to 2.5 × 10^4^ pfu was observed, as well as no cross-reactivity with the N proteins belonging to SARS-CoV-2, MERS-CoV, influenza virus, or the negative control on the nasal swab specimens.

To overcome the limitations of antibody production involving animal use, Raziq et al. developed a molecularly imprinted polymer-based electrochemical sensor for detecting the SARS-CoV-2 N protein (Figure 3C) [35]. The MIP sensor was prepared by modifying Micrux gold-based thin-film electrodes with a film generated from poly-m-phenylenediamine as a suitable functional monomer. Differential pulse voltammetry was used with ferro/ferricyanide as a redox probe for detecting N protein up to 111 fM, with LOD and LOQ values equal to 15 and 50 fM (0.7–2.2 pg/mL), respectively. To evaluate the matrix effect, the MIP sensor was tested in negative specimens by adding known concentrations of the N protein, with slightly higher values observed for the LOD and LOQ (27 fM and 90 fM, respectively). When tested with positive samples, good agreement was found between RT-PCR and the MIP sensor, demonstrating the feasibility of the developed MIP sensor.

## 3. SARS-CoV-2 Antigen Detection in Saliva

As highlighted above, nasopharyngeal swabs provide the main collection specimens, despite the procedure resulting in potential discomfort and requiring skilled healthcare personnel. Although underestimated in the first phase of the pandemic event as a specimen, saliva contains a detectable concentration of the virus and can be safely collected without the need for trained staff. To et al. reported that the salivary viral load was highest during the first week after symptom onset and subsequently declined over time [36]. Teo et al. collected saliva, nasopharyngeal swabs, and self-administered nasal swabs from 200 patients with acute respiratory infections and tested the samples via RT-PCR [37]. In total, 62.0%, 44.5%, and 37.7% of the saliva, nasopharyngeal, and self-administered nasal swabs gave positive results, highlighting that saliva represents a sensitive and suitable sample type for COVID-19 diagnosis.

For SARS-CoV-2 detection in saliva, we recently developed an electrochemical printed chip for the highly sensitive and accurate detection of SARS-CoV-2 in saliva. Since sensitivity and accuracy are key issues, we designed this immunosensor by employing: (i) magnetic beads as support for the immunological chain due to their ability to detect virus pre-concentration, thereby improving sensitivity; (ii) electrochemical detection, which is well-known as a sensitive and cost-effective detection method that uses a hand-held device; and (iii) carbon black as a nanomaterial to modify screen-printed sensors, thereby improving sensitivity in detecting the enzymatic by-product 1-naphtol and representing a cheap nanomaterial (around EUR 1 for 1 Kg). Furthermore, this device was conceived as an easy-to-use system. Thus, all the reagents needed for immunological chain creation can be added in a single step in untreated saliva, and during the incubation period of 30 min, stirring can be avoided (Figure 4A) [38]. To detect both N and S proteins, the antibodies for each antigen were selected and immobilized on magnetic beads. The two immunosensors developed were tested in untreated saliva, obtaining a detection limit equal to 19 ng/mL and 8 ng/mL, respectively, for S and N proteins. The effectiveness of these sensors was assessed using virus cultured in a biosafety-level-3 laboratory and in clinical samples from saliva for comparison against data obtained from nasopharyngeal swab specimens tested using Real-Time PCR. The immunosensor for S protein demonstrated higher sensitivity than the assay for N protein, with the former being able to measure 6.5 PFU/mL due to the high amount of S protein in SARS-CoV-2. Furthermore, when tested with saliva specimens, both immunosensors for N and S proteins were able, in almost all cases, to identify COVID-19 patient samples, even in the case of high CT values from Real-Time PCR (low viral load), demonstrating the high sensitivity of this cost-effective and miniaturized device.

Subsequently, Hall’s group used magnetic beads to develop a cheap (USD 3.20/test) aptamer assay for the detection of SARS-CoV-2 antigens using an off-the-shelf glucometer. In this study, a SARS-CoV-2 N- or S-protein-specific biotinylated aptamer was conjugated to a streptavidin-coated magnetic bead and pre-hybridized with a complementary antisense oligonucleotide strand covalently bonded to the invertase enzyme (Figure 4B) [39]. The analytical system is based on measuring the viral antigen’s interactions with the aptamer by quantifying invertase-antisense oligo release. The aptamer-antigen complex on magnetic beads was removed with a magnet, and the remaining aptamer-antisense-invertase complex was collected and incubated with sucrose, which was then converted to glucose and measured using the glucometer. To verify the effectiveness of the S and N aptamer-antisense-invertase system in detecting authentic virus and the native proteins produced during SARS-CoV-2 infection, the authors created viral stocks of SARS-CoV-2 in a biosafety-level-3 laboratory, demonstrating that the aptamer-antisense-invertase systems can recognize their native targets when produced by replicating authentic SARS-CoV-2. Subsequently, this aptasensor was challenged in saliva samples, showing a detection limit in saliva equal to 5.27 and 6.31 pM for N and S proteins, respectively. Finally, the authors tested the developed assays to discriminate SARS-CoV-2-infected and healthy individuals using validated saliva samples, with 100% positive percent agreement and 100% negative agreement with the RT-qPCR data performed on the same samples analyzed, demonstrating high accuracy and speed combined with cost-effectiveness.

Kelly et al. developed a faster reagent-free electrochemical immunosensor able to directly read out viral particles in 5 min using a sensor-modified electrode chip, without the addition of any reagents [40]. The sensor was built from an analyte-recognizing antibody attached to a rigid, negatively charged linker of DNA labeled with a ferrocene redox probe to produce the electrochemical signal. In this system, the application of a positive potential attracts the negatively charged DNA-labeled linker to the surface, thus producing a current. Because the drag force is affected by the size of the bound analyte, in the presence of the S protein (and better in the case of the virus), the response is changed. This sensing tool was successfully tested with saliva samples inactivated by heating at 65 °C for 30 min, and the results were found to be comparable with gold-standard RT-PCR approaches, demonstrating the effectiveness of this smart device (Figure 4C).

Gao et al. [41] developed a highly innovative electrochemical biosensing system called SARS-CoV-2 RapidPlex, which is characterized by multiple abilities, portability, and wireless connection for the detection of N protein, IgM and IgG antibodies, and inflammatory biomarker C-reactive protein using the same hand-held device (Figure 4D). This sensing tool is composed of mass-producible laser-engraved four-channel graphene arrays combined with a PCB system for wireless data transmission to a mobile user interface. For selective detection, the platform was chemically modified with captured antigens and antibodies to detect the target analytes. To assess the effectiveness of the device, N protein, IgM and IgG antibodies, and inflammatory biomarker C-reactive protein were analyzed in commercial saliva samples from RT-PCR COVID-19-confirmed patients (*n* = 5) and healthy subjects (*n* = 3). Using this device, the analysis required saliva-sample dilution in a phosphate buffer, followed by incubation for 10 min at room temperature, a washing step with the PBS buffer, and addition of the necessary reagents for 5 min. The results demonstrated the suitability of this smart device for multiplexing detection in saliva and serum samples, paving the way toward a highly innovative Telemedicine Platform for COVID-19 management.

## 4. SARS-CoV-2 Antigen Detection in Serum

The effectiveness of serum as a specimen for the detection of SARS-CoV-2 antigens has been also evaluated. For instance, Li et al. analyzed fifty cases of SARS-CoV-2 nucleic acid-positive and SARS-CoV-2 antibody-negative patients, observing an N protein positivity rate of 76%, suggesting that the serum measurement of SARS-CoV-2 N protein can have high diagnostic value for infected patients before the antibody appears, thus shortening the window of serological diagnosis [42]. Li and Lillehoj [43] developed a microfluidic device for high sensitivity measurements of SARS-CoV-2 N protein in undiluted and 5× diluted serum (Figure 5A). The printed electrochemical sensor was embedded in a microfluidic device able to minimize sample (25 μL) and reagent (80 μL) consumption and simplify handling of the sample. The sample was previously mixed with dually-labeled magnetic nanobeads (the beads were coated with a detection antibody and enzyme to improve signal amplification) and then dispensed into the chip using a capillary tube and plunger. Subsequently, the microfluidic chip was placed onto a magnet for 1 min to pre-concentrate the magnetic beads. Then, an incubation time of 50 min, in the case of whole serum samples, or 25 min, in the case of diluted serum samples, was selected. Subsequently, a buffer solution was flushed through the chip for 4 min at 100 μL/min, followed by the addition of an enzymatic substrate for 1 min at 100 μL/min for electrochemical measurements. The reported LOD of this immunosensor for SARS-CoV-2 N protein in the whole serum and 5× diluted serum was 50 and 10 pg/mL, respectively. To evaluate the effectiveness of the developed microfluidic device, the device was tested using serum samples obtained from seven COVID-19 patients and four healthy patients, observing a very low current (<1 μA) in the case of healthy people and a current in the range of 5–17 μA for COVID-19 patients, demonstrating results consistent with the PCR method.

## 5. SARS-CoV-2 Antigen Detection in Droplets

The mucosalivary droplets formed during breathing, speaking, coughing, and sneezing are the principal avenue by which people are infected. As described by Bourouiba [44], mucosalivary droplets are primarily composed of a multiphase turbulent gas cloud, which is unable to evaporate for a much longer time than isolated droplets. Mucosalivary droplets are thus characterized by a longer lifetime (by a factor of up to 1000), increasing the time of possible transmission from seconds to minutes. This behavior, unfortunately, increases the ease of infection among people that come into contact with such droplets in the absence of correctly worn masks.

A recent novel approach for evaluating an individual’s infection considers a mask not only as a protection system able to cut the diffusion of droplets [45] but also as a sampling and detection system. Several groups have sought to develop smart masks able to provide information on COVID-19 infection. For instance, Marrocco et al. developed a sensing face mask integrated with a radio frequency identification (RFID) tag for humidity sensing to monitor the wetness of the mask [46].

The only completed sensor embedded in a face mask for antigen detection was reported by Xue et al. [47]. These authors were the first to develop an immunosensor for the detection of SARS-CoV-2 in droplets by exploiting the surface of the face mask to collect and enrich the respiratory droplets (Figure 5B). This intelligent mask includes an impedimetric label-free immunosensor, a miniaturized impedance circuit including an A/D converter, an operational amplifier, and a wireless transmission unit. The authors developed the immunosensor to immobilize the antibodies for S protein on nanowires doped with biotin groups and thus achieve immobilization through streptavidin–biotin interactions. The nanowire array was designed with parallel patterns on the substrates and vertically connected via gold interdigitated electrodes. The width and spacing of the nanowires were set at 75 nm to increase their sensitivity for the detection of SARS-CoV-2 in aerosols, taking into account that high-density nanowire arrays allow for higher collision frequency between the immobilized antibodies and virus present in the droplets. The developed sensing system was able to detect the S protein and whole virus in simulated human breath, with a detection limit as low as 7 pfu/mL from an atomized sample of a coronavirus aerosol mimic and a measurement time of only 5 min.

## 6. Biosensors in the Literature vs. Commercialized Toolkits

As reported in the literature [16], ideal biosensors for deployment in pandemics should be simple to use. The ability to carry out measurements simply and effectively refers to both the detection of the target analyte and the sample treatment. Among commercially available kits for antigen detection, analytical tools are characterized by the collection of nasopharyngeal swabs, followed by simple suspension in a viral/universal transport medium and the addition of some drops on the strips. The response usually occurs in few minutes, with a range from 5 to 20 min (Table 1).

In commercial kits, unmeasured SARS-CoV-2 is largely caused by the uncorrected sampling of nasopharyngeal swab specimens. As reported by Lippi et al. for RT-PCR analyses based on nasopharyngeal swabs [48], this type of specimen sampling is characterized by preanalytical issues, such as: identification problems; inadequate procedures for the collection, handling, transport, and storage of the swabs; the collection of inappropriate or inadequate materials (in terms of quality or volume); and manual errors. Some analytical problems may also jeopardize diagnostic accuracy, including testing outside the diagnostic window, active viral recombination, the use of inadequately validated assays, insufficient harmonization, instrument malfunctions, and other specific technical issues. Moreover, this type of sampling is invasive, and its sensitivity is not sufficient to diagnose COVID-19 when a low load virus is present, which means high CT values.

The most significant advantage of the biosensors in the literature is their ability to work with other types of specimens—especially saliva, which is not invasive and does not require any sample treatment, as highlighted in Table 2. Indeed, both nasopharyngeal swabs and serum require complex sampling and/or sample treatments. In the case of serum, a laboratory-set-up treatment is required, which precludes the application of biosensors developed on-site [43]. To boost the applicability of saliva-based biosensors, which are less invasive and still highly sensitive, and enable them to reach market, several challenges remain to be addressed, including scalable manufacturing and storage stability, which are primary issues for any successful commercially available point-of-care device.

## 7. Conclusions

From the start of the present pandemic, all disciplines have made efforts to deliver useful tools to assist in the management of the outbreak. For on-site antigen detection, the industrial sector has combined established sampling techniques using nasopharyngeal swabs with customized lateral-flow systems and replaced previously used antibodies with other analytes and antibodies for S protein or N protein detection, with minimal variation in industrial-scale fabrication. Different companies have commercialized these types of devices. However, even when widely characterized by sufficient selectivity and sensitivity, these devices are not able to diagnose the infection at its start, at which point patients are characterized by a low viral load. To address this issue, researchers have begun to develop more sensitive devices by exploiting nanomaterial technology, microfluidics, and smartphone-assisted systems (Table 3). Nanomaterials, such as graphene and carbon black, have been used to increase the sensitivity of biosensors, and microfluidics combined with printed electrodes have facilitated the simple management of samples. Moreover, the presence of miniaturized potentiostats (e.g., Sensit Smart, PalmSens instrument) embedded in smartphones will help foster the synergic combination of biosensing tools with the Internet of Things. Indeed, the convergence of interdisciplinary technologies presents an immediate solution for the main bottlenecks that constrain biosensor prototypes from achieving real applications, matching the suitable features for effective use of a biosensor in pandemics, including long shelf life, ease of use, cost-effectiveness, mass manufacturing, autonomy, high sensitivity, high selectivity, rapidity, multiplexing capabilities, multiple sensing modes, and single-use [16].

However, several main drawbacks still need to be overcome, as highlighted by Eden Morales-Narváez and Can Dincer [12]. Previous recommendations include the need for: (i) investments in diagnostic tools; (ii) collaborative networks within the biosecurity sector; (iii) autonomy for each country to manufacture its own biosensing technologies and protection equipment; (iv) the development of new diagnostic tools to meet government requirements; (v) widely establishing the necessary features of biosensors; (vi) the connection of biosensors with the Internet of Things; and (vii) training of the population for self-sampling and testing.

Taking into account the results achieved to date as well as the recommendations for the successful use of biosensors in virus detection, we are confident that this pandemic event will positively affect biosensing research activity. Biosensors with all the features required for reliable applications should be developed with the overriding goal of producing devices not only confined in articles but also able to be used among the population; this point was highlighted by Mao et al. [49] when discussing “the feasibility of an integrated point-of-care biosensor system with mobile health for wastewater-based epidemiology (iBMW) for early warning of COVID-19, screening and diagnosis of potential infectors, and improving health care and public health”. Thus, beyond scientific publications, the delivery of useful toolkits in collaboration with companies should be one of the main goals for the scientific community, in order to avoid the monitoring issues observed in the COVID-19 pandemic during a future unexpected outbreak.

## Figures and Tables

**Figure 1 biosensors-11-00310-f001:**
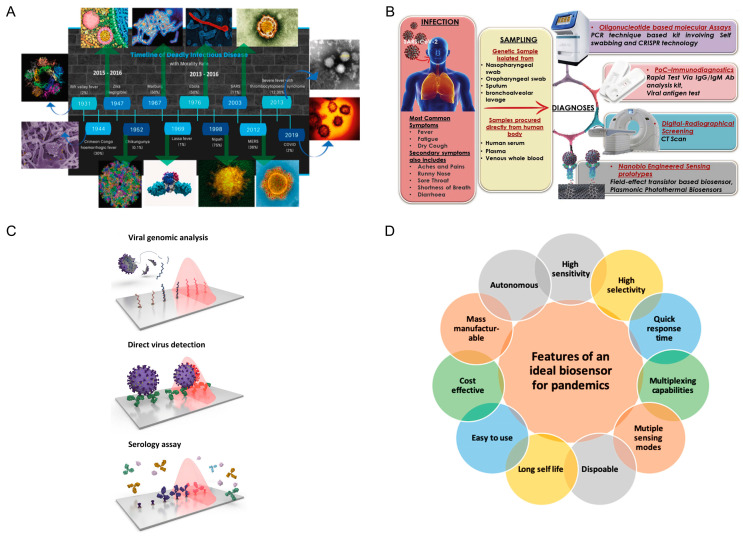
(**A**) Timeline of virus-based diseases. Reprinted with permission from [1], 2021 Elsevier. (**B**) different diagnostic methods for COVID-19. Reprinted with permission from [4], 2020 Elsevier. (**C**) different strategies for biosensing tool development in the COVID-19 outbreak. Reprinted with permission from [5], 2020 American Chemical Society and (**D**) features of an ideal biosensor for pandemics. Reprinted with permission from [16], 2020 American Chemical Society.

**Figure 2 biosensors-11-00310-f002:**
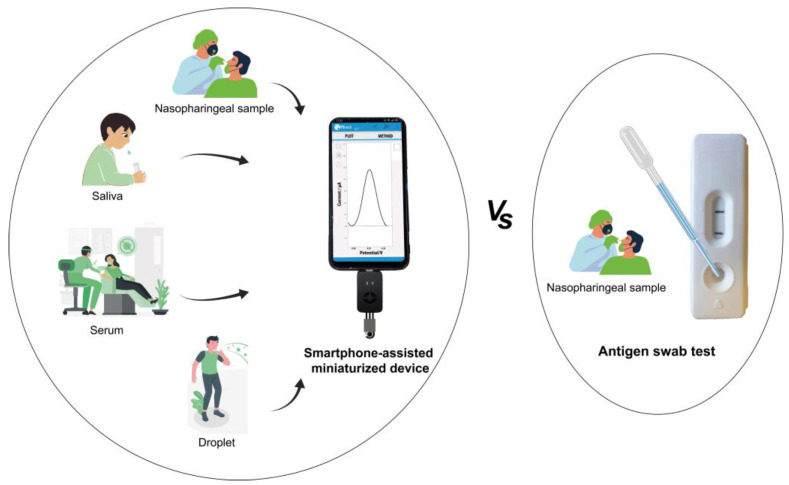
Different matrices used for the development of antigen biosensors at an academic level vs. commercially available antigen kits.

**Figure 3 biosensors-11-00310-f003:**
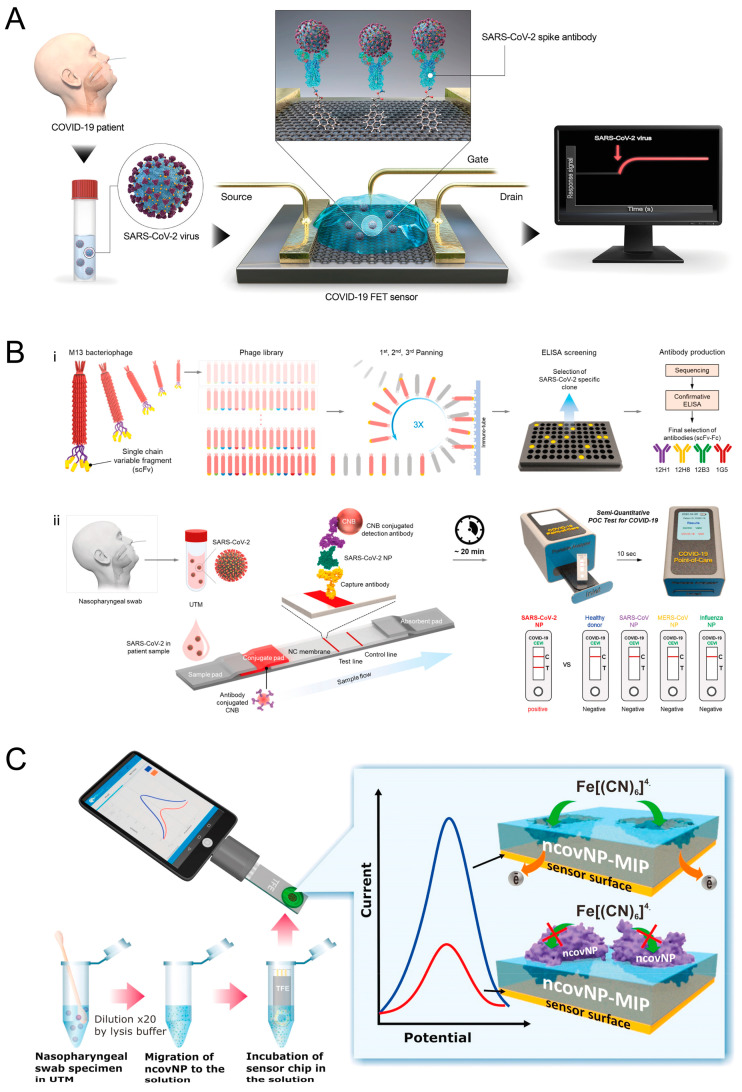
SARS-CoV-2 antigen detection in a nasopharyngeal swab. (**A**) S protein detection with a field-effect transistor-based immunosensor. Reprinted with permission from [26], 2020 American Chemical Society; (**B**) lateral-flow assay fabricated using the scFv-Fc antibody for N protein detection. Reprinted with permission from [34], 2021 Elsevier; (**C**) molecularly imprinted polymer-based electrochemical sensor for the detection of N protein. Reprinted with permission from [35], 2021 Elsevier.

**Figure 4 biosensors-11-00310-f004:**
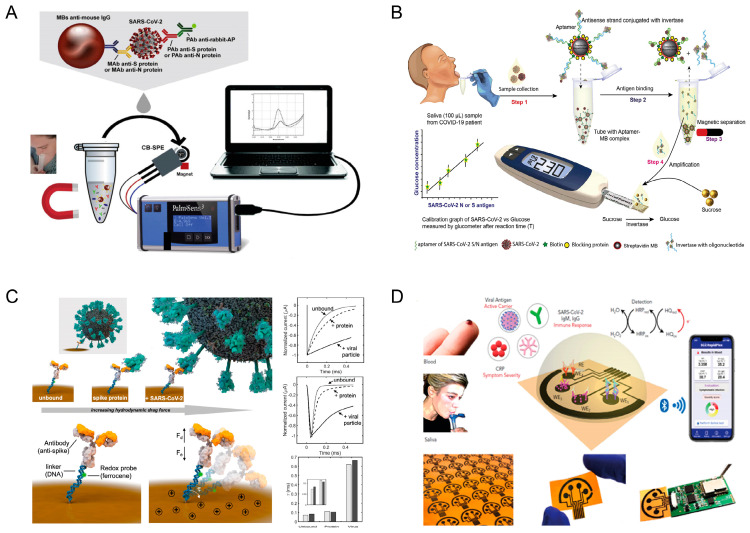
SARS-CoV-2 antigen detection in saliva: (**A**) magnetic beads combined with a nanomaterial-based-printed electrode for the development of two immunosensors for the detection of S and N proteins. Reprinted with permission from [38], 2021 Elsevier; (**B**) magnetic beads for the development of a cheap aptamer assay for the detection of S and N protein antigens, exploiting an off-the-shelf glucometer. Reprinted with permission from [39], 2021 Elsevier; (**C**) a reagent-free electrochemical immunosensor for directly reading out viral particles in 5 min. Reprinted with permission from [40], 2020 American Chemical Society; (**D**) an electrochemical biosensing system for the detection of N protein, IgM and IgG antibodies, and inflammatory biomarker C-reactive protein using the same hand-held device. Reprinted with permission from [41], 2020 Elsevier.

**Figure 5 biosensors-11-00310-f005:**
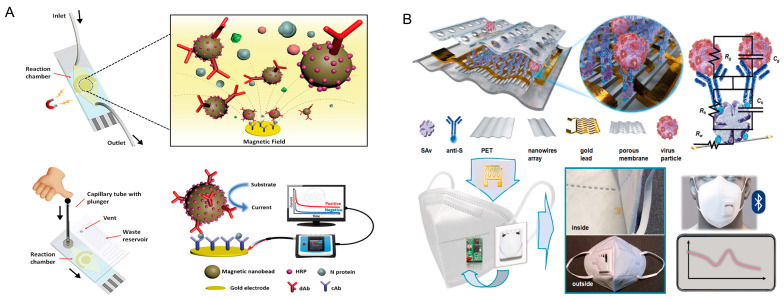
SARS-CoV-2 antigen detection in serum and droplets. (**A**) Microfluidic device for the high-sensitivity measurement of N protein in undiluted and 5× diluted serum. Reprinted with permission from [43], 2021 American Chemical Society; (**B**) immunosensor for the detection of SARS-CoV-2 in droplets by exploiting the surface of the face mask to collect and enrich the respiratory droplets. Reprinted with permission from [47], 2021 Elsevier.

**Table 1 biosensors-11-00310-t001:** Commercially available antigen kits.

Antigen Swab Test	Sensitivity %	Specificity %	Limit of Detection (TCID_50_/mL)	Analysis Time (Min)	Ref.
STRONGSTEP	96	99	2.50 × 10^2^	15	[18]
BIOCREDIT	90	90	Not reported	5–8	[19]
REALY TECH	90	100	1.25 × 10^3^	10–20	[20]
VIVADIAG	83	100	1.35 × 10^3^	15	[21]
ZKDENTAL	87	100	Not reported	15	[22]
MOLAB	99	98	1.15 × 10^2^	15	[23]
JOYSBIO	89	99	1.60 × 10^2^	15	[24]
CLUNGENE	91	100	5 × 10^2.67^	15	[25]

**Table 2 biosensors-11-00310-t002:** Sample treatment of biosensors in the literature.

Sample Matrix	Treatment	Ref.
Nasopharyngeal swab	Nasopharyngeal swabs were suspended in a universal transport medium	[26]
Nasopharyngeal swab	Nasopharyngeal swabs were suspended in a viral transport medium	[32]
Nasopharyngeal and throat swabs	Nasopharyngeal and throat swabs were mixed in a viral transport medium	[33]
Nasopharyngeal swab	Not reported	[34]
Nasopharyngeal swab	Nasopharyngeal specimens were vortexed in a universal transport medium	[35]
Saliva	No treatment	[38]
Saliva	No treatment	[39]
Saliva	No treatment	[40]
Saliva	No treatment	[41]
Serum	Whole and 5× diluted	[43]
Droplets	No treatment	[47]

**Table 3 biosensors-11-00310-t003:** Analytical features of biosensors for the detection of N and S proteins reported in the literature.

Analyte	Type of Biosensor	Type of Transduction	Matrix Analyzed	Linear Range (LR)/Detection Limit (LOD)	Time of Analysis	Ref.
S protein/virus	Graphene-based field-effect transistor immunosensor	FET	Nasopharyngeal swab	S protein LR: 0.1–100 pg/mL LOD: 100 fg/mL Virus: LR: 1.6 × 10^1^–1.6 × 10^4^ pfu/mL LOD: 1.6 × 10^1^ pfu/mL	<1 min	[18]
S/N protein	Single-walled carbon nanotube-based field-effect transistor immunosensor	FET	Nasopharyngeal swab	S protein LR: 5.5 fg/mL–5.5 pg/mL LOD: 0.55 fg/mL N protein LR: 16 fg/mL–16 pg/mL LOD: 0.016 fg/mL	<5 min	[24]
N protein	Lateral-flow immunoassay	Colorimetric (visual) detection	Nasopharyngeal swab	sensitivity 98.33%	15–30 min	[25]
N protein/virus	Lateral-flow immunoassay based on scFv-Fc fusion proteins	Colorimetric detection	Nasopharyngeal swab	N protein LOD: 2 ng Virus LOD: 2.5 × 10^4^ pfu	20 min	[26]
N protein	Molecularly imprinted polymer-based electrochemical sensor	Differential Pulse Voltammetry	Nasopharyngeal swab	LOD: 27 fM in clinical samples	15 min	[27]
S/N protein and virus	Magnetic bead-based immunosensor combined with carbon black-modified screen-printed electrode	Differential Pulse Voltammetry	Saliva	S protein LOD: 19 ng/mL in saliva N protein LOD: 8 ng/mL in saliva Virus: 6.5 pfu/mL concentration tested using S protein immunosensor and 6.5 × 10^3^ pfu/mL concentration tested using N protein immunosensor	30 min	[30]
S/N protein	Magnetic bead-based sensor using a biotinylated aptamer-oligo-invertase complex	Glucometer	Saliva	N protein LOD: 5.27 pM in saliva S protein LOD: 6.31 pM in saliva	60 min	[31]
S protein/virus	A reagent-free electrochemical Sensor modified with an antibody attached to DNA linker functionalized with a ferrocene redox probe	Chronoamperometry	Saliva	S protein LOD: 1 pg/mL Virus LOD: 4000 copies per mL	5 min/10 min	[32]
N protein	An electrochemical immunosensing platform using laser-engraved graphene electrodes and a wireless transmission unit	Chronoamperometry	Saliva	LR: Up to 5000 pg/mL	1 min	[33]
N protein	Microfluidic immunosensor based on screen-printed gold electrode combined with magnetic nanobeads	Chronoamperometry	Serum	Whole serum LOD: 50 pg/mL 5× Diluted serum LOD: 10 pg/mL	Whole serum: 50 min 5× diluted serum: 25 min	[35]
S protein/virus	Nanowire-based immunosensor combined with a miniaturized impedance circuit and a wireless transmission unit	Electrochemical impedance spectroscopy	Droplets	S protein in aerosol concentration tested: 1, 10 ng/mL Virus aerosol LOD: 7 pfu/mL corresponding to an air concentration of 0.35 pfu/L	S protein aerosol: 10 min Virus aerosol: 5 min	[39]

## Data Availability

Not applicable.

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
