# Peer review of "State of the Art on the SARS-CoV-2 Toolkit for Antigen Detection: One Year Later"

_biosensors, 2021, doi:10.3390/bios11090310_

Round 1

Reviewer 1 Report

The authors have written a review article about “State of the art on SARS-CoV-2 Toolkit for antigen detection: 1-year after:” for Rapid tests which are strongly required for antigen detection for easy and quick evaluation of contagious people, with the overriding goal to delimitate the virus spread among the population. The manuscript is well structured and it is of special interest towards biosensors community. The information included in the tables are also quite interesting. However, I have some issues regarding this review. These issues should be addressed.

  1. This review is just a summary rather than the authors own comments and suggestions etc, what are the authors own comments, suggestion and what are the future prospects and challenges for graphene society? How this article plays an effective role? This review looks like a report rather than a review. Authors should give their own opinion depending upon their expertise in each section regrinding the title.
  2. The title is clear that is adequate to the content of the article.
  3. High resolution figures should be placed for easy understanding. E.g. the text is unclear in almost all figures. I will recommend TIFF file.
  4. As all the figures were taken from other source, it is important to get the copyrights permission before publication etc.
  5. I will also recommend the following articles to cite as they are recent published articles regarding COVID-19 detection. (i) https://doi.org/10.1186/s12985-020-01452-5 (ii) https://doi.org/10.1016/j.bios.2021.113005 After corrections and additions, the manuscript can be published in this journal

Reviewer 2 Report

This review concluded SARS-CoV-2 toolkits for antigen detection. It is a timely and good review work. It suggests organizing the section depending on toolkit formats. The original sections (2. SARS-CoV-2 antigen detection in nasopharyngeal swab, 3. SARS-CoV-2 antigen detection in saliva, 4. SARS-CoV-2 antigen detection in serum, 5. SARS-CoV-2 antigen detection in droplets) could confuse readers due to a sizeable cross-covered content. It could be further considered after re-arranging its sections.

  1. The main point of the review should be listed here.
  2. Please make it clear why choosing antigen detection instead of antibody detection.
  3. Please include a section of sample preparation.
  4. Please re-organize the work depending on the SARS-CoV-2 toolkit format. For example, test strips, microfluidics, or smartphone.
  5. Please add a section to evaluate commercialized toolkits

Author Response

Please see attachament
